# Genetic Alterations of Transcription Factors and Signaling Molecules Involved in the Development of Congenital Heart Defects—A Narrative Review

**DOI:** 10.3390/children10050812

**Published:** 2023-04-29

**Authors:** Alexandru Cristian Bolunduț, Cecilia Lazea, Carmen Mihaela Mihu

**Affiliations:** 11st Department of Pediatrics, “Iuliu Hațieganu” University of Medicine and Pharmacy, 400370 Cluj-Napoca, Romania; 21st Pediatrics Clinic, Emergency Pediatric Hospital, 400370 Cluj-Napoca, Romania; 3Department of Histology, “Iuliu Hațieganu” University of Medicine and Pharmacy, 400012 Cluj-Napoca, Romania

**Keywords:** congenital heart defects, TBX5, GATA4, NKX2-5, CRELD1

## Abstract

Congenital heart defects (CHD) are the most common congenital abnormality, with an overall global birth prevalence of 9.41 per 1000 live births. The etiology of CHDs is complex and still poorly understood. Environmental factors account for about 10% of all cases, while the rest are likely explained by a genetic component that is still under intense research. Transcription factors and signaling molecules are promising candidates for studies regarding the genetic burden of CHDs. The present narrative review provides an overview of the current knowledge regarding some of the genetic mechanisms involved in the embryological development of the cardiovascular system. In addition, we reviewed the association between the genetic variation in transcription factors and signaling molecules involved in heart development, including TBX5, GATA4, NKX2-5 and CRELD1, and congenital heart defects, providing insight into the complex pathogenesis of this heterogeneous group of diseases. Further research is needed in order to uncover their downstream targets and the complex network of interactions with non-genetic risk factors for a better molecular–phenotype correlation.

## 1. Introduction

Congenital heart defects (CHD) can be defined as structural abnormalities of the heart and/or the great vessels present at birth and are caused by alterations in the normal embryological development of the cardiovascular system. They are a heterogeneous group of diseases. Botto et al. proposed a classification for etiologic studies based on anatomical, clinical and embryological similarity [1]. They divided CHDs into eight categories: conotruncal defects (including truncus arteriosus, the interrupted aortic arch, D-transposition of the great arteries, tetralogy of Fallot and the double outlet right ventricle), atrioventricular septal defect, anomalous-pulmonary venous return (total or partial), left ventricular outflow tract obstruction (including hypoplastic left heart syndrome, coarctation of aorta and aortic stenosis), right ventricular outflow tract obstruction (including pulmonary valve stenosis, tricuspid atresia, Ebstein malformation and pulmonary atresia), septal defects (ventricular or atrial), heterotaxy malformations, and complex defects (including the L-transposition of the great arteries, univentricular heart and other associations) [1].

CHDs are the most common congenital abnormality, followed by limb anomalies, congenital anomalies of the kidney and urinary tract and nervous system anomalies [2]. The overall global birth prevalence of CHDs reported was 9.41 per 1000 live births [3], which shows a substantial increase in recent decades [3,4]. The prevalence showed marked differences between different geographical areas [3,4]. The most common type is the ventricular septal defect (VSD), representing about 35% of all CHDs, followed by the atrial septal defect (ASD) and patent ductus arteriosus (PDA) [3]. The prevalence of ‘mild’ lesions (defects that usually have a less significant hemodynamic and clinical impact, including VSD, ASD and PDA) increased during the last decades, probably due to advances that led to more accurate detection, while the ‘severe’ lesions showed a relatively constant trend in prevalence [3].

The etiology of CHDs is complex and still poorly understood. Clearly identified environmental factors (non-genetic risk factors) account for about 10% of CHDs [5]. The most studied of them include maternal health conditions (such as obesity and diabetes mellitus), maternal alcohol consumption, maternal smoking, maternal exposure to certain drugs during pregnancy (antidepressants, anticonvulsants and antiarrhythmic drugs), maternal fever during the first trimester (which possibly correlates with the teratogenic effect of different microorganisms), paternal smoking and paternal advanced age [6,7]. Genetic contributions represent the remaining 90% [5]. Table 1 summarizes some of the current knowledge about the etiology of CHDs. In a review regarding their complex inheritance, Diab et al. concluded that more than half of CHDs do not have an identified etiology, which will probably, in time, be explained by gene-to-gene interactions, gene-to-environment interactions, polygenic inheritance or epigenetic mechanisms, all of which are still under intense research [5]. New studies based on single-cell RNA-sequencing and other functional genetics methods are starting to uncover the network of transcription factors and signaling molecules involved [8,9]. Whole-genome sequencing studies have underlined the implication of non-coding de novo mutations as part of the complex genetic model of CHDs, but their effect on the transcriptional activity of coding genes and their burden on the alteration of heart development is still unclear [10].

The aim of this narrative review is to provide an overview of the current knowledge regarding some of the genetic mechanisms involved in the embryological development of the cardiovascular system. We briefly summarized the embryologic stages of heart development and overviewed the role played by some transcription factors and signaling molecules in this process. We also reviewed their association with congenital heart defects, providing insight into the complex pathogenesis of this heterogeneous group of diseases.

## 2. Embryologic Stages of Heart Development

### 2.1. Formation of the Heart Tube

The heart is one of the first organs to develop in the human embryo. It originates mostly from the mesoderm, with the contribution of ectodermal-derived cells (neural crest cells) [11]. The heart-forming region (the first heart field) is composed by the first mesodermal cells that migrate through the primitive streak and position themselves in a horseshoe-like shape, rostral and lateral to the stomato-pharyngeal membrane [11,12,13]. During this process, they exit the region of blocked differentiation mediated by *Wnt* growth factors and enter an environment that promotes cardiomyocyte differentiation, via active *Wnt* inhibition; the expression of bone morphogenetic protein (*BMP*) growth factors; and other cardiac-specific transcription factors, such as *Nkx2.5* and *Islet1* [12,14,15,16].

The closure of the neural tube and the development of the nervous system determines the cranio-caudal and anterior folding of the embryo, which places the heart-forming region in its final position [11]. This process leads to the formation of the heart tube, a Y-shaped structure with two caudo-lateral inlet branches and one cranial outlet, which is initially attached to the body, anterior to the foregut, by the dorsal mesocardium [17] and organized in three layers: an outer layer of cardiomyocytes (termed primary myocardium), an inner layer of endocardial cells (a cell population that is also derived from the splanchnic mesoderm of the heart field at the same time as cardiomyocytes [11,18]) and an extracellular matrix between them (the cardiac jelly) [12]. After that, the heart tube starts to loop towards the right, in a progressive manner, forming a C-like and then an S-like structure [12,17]. The looping is mediated by the growth of the heart tube due to the addition of new cardiomyocytes, derived from the second heart field (located posteriorly from the first heart field), and migrated through the dorsal mesocardium [11,19,20].

The cellular ultrastructure of cardiomyocytes makes all of them capable of contraction (via sarcomeres), spontaneous depolarization and the conduction of the electrical impulses generated (via ion pumps or channels and gap junctions) [11,12]. Cardiomyocytes in various regions of the heart differ from one another by the intensity of these characteristics and are partly mediated by the expression of the T-Box (*TBX*) transcription factor family [11]. The fate of one cell is determined by its location in the developing heart via signals that promote the expression of certain transcription factors and other molecules [11].

### 2.2. Development of Heart Chambers

The looping of the heart tube represents a scaffold for the future four-chambered heart. During this process, the cardiac jelly between the endocardium and the myocardium disappears at the outer curvatures so the cardiomyocytes in these regions resume proliferation, along with differentiation, forming the primordia for the future ventricles at the arterial pole and the atrial appendices at the venous pole. This is referred to as the ballooning model [11]. The proliferating and differentiating cardiomyocytes are termed secondary or chamber myocardium and express specific genes such as the atrial natriuretic factor (*Anf*/*Nppa*) and Connexin 40 (*Cx40*/*Gja5*), a gap-junction protein [11,21]. The cardiomyocytes form endoluminal protrusions, called trabeculations, that grow via the addition of new cells at their base and assure the lengthening and thickening of the myocardium [12,20]. The initial ventricles have a trabecular phenotype. During heart maturation, under the influence of epicardial-derived fibroblasts, a compact layer develops, with a different set of markers expressed compared to the trabecular myocardium [22].

The secondary myocardium gives rise only to the trabeculated portion of the mature atria. The rest of the atrial myocardium forms via the incorporation of the venous system [11]. The systemic venous blood is retrieved to the heart by the left and right common cardinal veins. The pericardial cavity expands and incorporates the common cardinal veins, while cardiomyocytes encircle them, forming the so-called sinus venosus, with its right horn becoming the dorsal part of the right atrium [11,12]. The pulmonary venous system develops as a vascular plexus surrounding the embryonic foregut, which gives rise to the lung primordia. This plexus drains through a single vessel that is connected to the atrium in the midline via the dorsal mesocardium. Cardiomyocytes encircle the pulmonary vein and incorporate it into the future left atrium up to the second bifurcation, resulting in four pulmonary orifices [11,12].

### 2.3. Septation of the Heart

The septation of the heart is a complex process that leads to the development of its closed double circulation characteristic. The process starts with the formation of extracellular matrix cushions in the atrioventricular canal and the outflow tract, which are further populated by endocardium-derived mesenchymal cells [12].

Atrial septation begins with the primary septum, a crescent-shaped structure formed by the proliferation of atrial cells, growing towards the atrioventricular canal and closing the primary atrial foramen (ostium primum) by fusing with the major atrioventricular cushions [11,12]. To ensure the bypass of the pulmonary circulation characteristic for the fetal circulation, a secondary atrial foramen (ostium secundum) is formed in the primary septum by the apoptosis of the cells [23,24]. To complete the septation process, the muscular wall folds down to the right side of the primary septum, forming the secondary septum and completely covering the ostium secundum. The secondary septum is still incomplete, and the orifice that remains forms with the ostium secundum, an oblique canal termed the foramen ovale, which represents the right-to-left shunt characteristic to the fetal circulation [23,24]. At birth, due to the change in atrial blood pressure, the primary and secondary septum are pushed together and the atrial septation is completed [11,23].

The septation of the ventricles starts during the ballooning process with the formation of the muscular septum, by apposition, adding cells at its base from the adjacent growing left ventricular free wall [12]. With the inner curvature, the superior rim of the ventricular septum determines the ventricular foramen, which is functionally separated by the major (dorsal and ventral) atrioventricular cushions in a right and left portion [12]. The ventricular septation will become anatomically complete with the septation of the outflow tract.

The outflow tract connects the ventricles to the aortic sac, which in turn is connected to the third, fourth and fifth pairs of pharyngeal arch arteries [12]. The septation of the outflow tract starts at the level of the cushions from this region, in a spiral fashion and from distal to proximal, generating a 180° turn in the future aortic and pulmonary arteries [11,12]. The outflow tract cushions are populated with cells migrated from the secondary heart field, as well as cells migrated from the neural crests, which provide growth factors essential for the normal development of the region [12,25]. Proximally, the outflow tract septum will fuse with the atrioventricular cushions, forming the membranous ventricular septum, completing the anatomical septation of the ventricles. Distally, a protrusion of the pharyngeal mesenchyme, termed the aorto-pulmonary septum, connects with the outflow tract cushions, separating the emerging arteries [12]. The system of pharyngeal arch arteries will remodel and generate the adult arterial pattern at the end of the eighth week of development [12]. The cardiac jelly at the site of the atrioventricular and outflow tract cushions and ridges will remodel, forming the heart valves, in a complex process [11].

Table 2 summarizes the milestones of heart development and their connections with CHDs.

## 3. An Overview of Transcription Factors and Signaling Molecules Involved in the Development of CHD

### 3.1. TBX5

*TBX5* is a transcription factor from the T-box family with a highly conserved expression pattern in the heart, forelimb and retina in different vertebrate species [26].

#### 3.1.1. *TBX5* in Heart Development

The T-box transcription factor family plays an important role in regulating the cardiomyocyte identity in the developing heart [11]. *TBX5* demonstrates an expression gradient from the venous pole of the heart (the region of the developing atria) to the right ventricle and the outflow tract, where it is absent [11,26]. Recent studies on human embryonic stem cell-derived cardiomyocytes uncovered both structural and functional roles that can be attributed to *TBX5* expression, which include upregulating sarcomere structure genes and genes involved in cellular calcium handling [27].

The transcription factor is involved in the septation of the heart, for both the atria and the ventricles. The location of the ventricular septum depends on the *TBX5*/*TBX20* interaction, while the boundary between the *TBX5* positive/*TBX20* negative cardiomyocytes (specific to the left ventricle) and the *TBX5* negative/*TBX20* positive myocardium (specific to the right ventricle) appears to indicate the position of the interventricular septum [28,29]. In addition, the development of the atrial septum is dependent on the expression of *TBX5* in the second heart field and is regulated by interactions with Sonic Hedgehog (*SHH*) signaling, with the *SHH* ligand secreted from the pulmonary endoderm [30,31].

#### 3.1.2. *TBX5* in the Development of the Ventricular Conduction System

*TBX5* appears to play a central role in the development of the ventricular conduction system via the specification and regulation of gene expression. It activates the expression pattern for the fast-conducting phenotype characteristic for the ventricular conduction system cells by driving the gene regulatory network, composed of genes such as *CX40*, *SCN5A*, *RYR2*, *KCNK3* or *KCNJ2,* and coding for Na^+^, K^+^ and Ca^++^ channels, both in the developing and the adult heart [26,32,33]. Van Ouwerkerk et al. proposed a murine model to evaluate the effects of a *TBX5* missense mutation (c.373G>A; p.Gly125Arg) in relation to early onset atrial fibrillation. The mice showed variable RR intervals and a susceptibility to atrial fibrillation, similarly to patients with the similar mutation. The cardiomyocytes demonstrated decreased systolic and diastolic intracellular calcium concentrations, thus affecting action potentials, probably due to changes in the regulatory element activity and transcriptional regulation (epigenetic changes) induced by the *TBX5* mutation, confirming the central role played by the transcription factor in the development of the conduction system [34].

#### 3.1.3. Gene–Environment Interactions Involving *TBX5*

Zhang et al. demonstrated that increased maternal leucine levels in the first trimester of pregnancy increase the risk for congenital heart defects in the offspring due to the inhibition of *TBX5* signaling. In an experimental model performed on mice, they showed that increasing the maternal leucine levels will increase embryonic lysine-leucylation, thus modifying the lysine residues in *TBX5* and inhibiting its transcriptional activity [35]. This underlines the importance of gene–environment interactions and the complex mechanisms involved in the pathogenesis of CHD.

#### 3.1.4. Genetic Variation Involving *TBX5*

*TBX5* haploinsufficiency has been shown to be the cause of Holt–Oram syndrome (HOS), an autosomal dominant disorder affecting cardiac and upper limb development, with a high penetrance and variable expression [36,37,38]. The cardiac phenotype consists of congenital heart defects (mostly septal defects, the atrial septal defect being the most frequent) and conduction abnormalities, while the characteristic limb anomalies are bilateral and asymmetric radial ray defects [39]. The majority of the reported *TBX5* mutations are nonsense or frameshift mutations, which result in the introduction of premature stop codons, obtaining a truncated protein with impaired function. Al-Qattan et al. reviewed the genotype–phenotype correlation in typical HOS patients resulting from missense mutations of *TBX5*, concluding that most of the mutations were found in the DNA-binding domain of the gene, thus affecting the interaction with other transcription factors regulating the embryological development of the heart and limbs (such as *GATA4* or *NKX2-5*) and causing loss of function [40].

Recent reports have shown a heterogeneous pattern of genetic variants of the *TBX5* gene in relation to heart disease, in addition to the known features of typical HOS, underlining the complex molecular regulation of the developing heart and uncovering new roles for this transcription factor during the embryological period. Table 3 summarizes some of the newly identified mutations involving *TBX5* and their phenotype.

### 3.2. GATA4

*GATA4* is a transcription factor from the *GATA* family characterized by a highly conserved two zinc fingers domain that binds to the specific DNA sequence 5′-(A/T)GATA(A/G)-3′ in the promoters of target genes [52].

#### 3.2.1. *GATA4* in Cardiomyocyte Differentiation

Many cardiac-specific genes possess GATA elements in their regulatory regions, including not only structural sarcomere genes, such as myosin light chain-3 (*MYL3*) or troponin C (*TNNC1*) and I (*TNNI3*), but also genes for Na^+^/Ca^2+^-exchanger (*SLC8A3*), acetylcholine receptor-M2 (*CHRM2*), cardiac-restricted ankyrine repeat protein (*CARP*), carnitine palmitoyltransferase-1b (*CPT1B*), and other transcription factors, such as *NKX2-5*, which have been shown to be regulated by *GATA4* [52]. Due to the overlap in the DNA-binding region, the cardiac subfamily of *GATA* transcription factors (*GATA4/5/6*) exhibit redundant functions and interact with each other to ensure the specification of cardiac cells and subsequent heart development [53].

Zhao et al. suggested that *GATA4* and *GATA6* are essential for heart development by influencing the transcription factors network involved in the cardiomyocyte differentiation. They demonstrated that *GATA4*/*GATA6* knockout mice presented with acardia, blocking the formation of the first heart field and the differentiation of the progenitor cells into cardiomyocytes, while the formation of the second heart field was independent of *GATA4* activity [54]. Various in vitro studies have underlined the central role of the transcription factor and its interactions with other regulatory elements in inducing cardiac specification. For example, the combination of *GATA4*, *NKX2-5* and *TBX5* was sufficient for the activation of the cardiac genetic program, both for transdifferentiating mesodermal cells and for reprogramming postnatal fibroblast to differentiated cardiomyocytes [55,56].

#### 3.2.2. *GATA4* in the Development of the Atrioventricular Region

*GATA4* has an important role in the development of the atrioventricular region, including the valvulo–septal complex. The endocardium and endocardial cushions have a high expression of the transcription factor, and it was shown that *GATA4* mediates both epithelial-to-mesenchymal transition (EMT) in the endocardial cushions and the growth and remodeling of the atrioventricular cushions, playing an important role in valve development [57]. Furthermore, *GATA4* interacts with different cofactors in order to promote cell-type-specific gene expression programs [53]. For example, Kim et al. demonstrated the interaction between *GATA4* and *RERE* (arginine-glutamic acid dipeptide repeats gene), a cardiac-expressed nuclear receptor co-regulator, in the development of the membranous portion of the ventricular septum, via their effects on EMT and mesenchymal cell proliferation [58]. Functional analyses of both *GATA4* and *FOG2*/*ZFPM2* gene mutations have shown a disturbance in the normal interaction between the two transcription factors, leading to similar phenotypes (mainly conotruncal defects), suggesting that the gene pair has an important role in the development of the outflow tract [59,60]. In addition, *GATA4* is involved in the localization of the atrioventricular canal by activating enhancers that lead to H3K27 acetylation, thus activating atrioventricular canal-specific gene loci. This is in contrast to its role in the chamber myocardium, where it determines deacetylation and gene repression [61,62]. In a murine model, *GATA4* haploinsufficiency determined an impaired migration of Hedgehog-responsive progenitor cells from the second heart field in the outflow tract, causing CHDs [63]. Further studies have shown that *GATA4*, in association with *NKX2-5*, promotes the maturation of differentiated cardiomyocytes after the migration [64].

#### 3.2.3. *GATA4* in the Developed Heart and Genetic Variation Involving *GATA4*

The transcription factor plays an important role in the adult heart as well by promoting the compensatory mechanisms determined by different types of acquired cardiovascular diseases, such as hypertrophy and angiogenesis. *GATA4* is upregulated by mechanical and neurohumoral stimuli, such as increased hemodynamic load, direct wall stretching, angiotensin II, and endothelin-1, thus activating hypertrophy-associated gene promoters [52,65].

Many *GATA4* mutations are associated with congenital heart defects. Table 4 summarizes some of the identified mutations involving the genes and their phenotypes.

### 3.3. NKX2-5

The transcription factor *NKX2-5* is part of the *NK-2* homeodomain-containing family, which is related to the *Drosophila* tinman protein (essential for the development of the dorsal vessel in flies) and highly conserved in many different species, from flies to zebrafish and humans [74,75,76,77]. The family is characterized by a homeodomain that consists of a helix-turn-helix motif and binds to the specific DNA sequence 5′-T(C/T)AAGTG-3′ [74,75].

#### 3.3.1. *NKX2-5* in Heart Development

During the embryologic period, *NKX2-5* is expressed in the mesodermal cells of the first heart field, as well as the second heart field and the endoderm underlying it, indicating a central role in the transcriptional regulation of the developing heart due to its expression in progenitor cells [75,76]. Tissue-specific deletions in mouse models have shown that the mesodermal expression of *NKX2-5* is essential for heart development, with a heart phenotype similar to that of *NKX2-5* knock-out mice in the mesodermal deletion group, while the endodermal expression has very little influence on the normal development of the heart [76].

George et al. revealed that *NKX2-5* expression is essential for the maintenance of the ventricular identity of cells during cardiomyocyte differentiation and acts in a dose-dependent manner in preserving the chamber-specific molecular features during atrial and ventricular formation [78]. In addition, studies have shown that *NKX2-5* plays a specific role in the development of the outflow tract and the right ventricle [76,77]. Immunohistochemistry studies have demonstrated an increased expression of *NKX2-5* in the outflow tract wall, thus promoting the progressive maturation of the differentiated cardiomyocytes at this level [64].

#### 3.3.2. *NKX2-5* in the Development of the Ventricular Conduction System

*NKX2-5* appears to be involved in the development of the conduction system by influencing the number of specific myocytes in a dose-dependent manner, with little effect on the transcriptional regulation of gap junction proteins [79]. Moreover, it directly activates the expression of several major cardiac ion-channels, such as *SCN5A*, *CACNA1C* and *KCNH2*, thus linking the gene with conduction disturbances [80]. Li et al. uncovered that *NKX2-5* is required for maintaining the physiological function of the sinoatrial node, with sinus node dysfunction as a result of the downregulation of the transcription factor, although it is not essential for the morphological development of the structure [81].

#### 3.3.3. Gene–Gene and Gene–Environment Interactions Involving *NKX2-5*

Various transcription factors and other regulatory elements interact with one another in order to mediate the normal development of the heart. For example, Vincentz et al. identified a direct interaction between *NKX2-5* and *MEF2C* for ventricular chamber formation by influencing ventricular cardiomyocyte specification [82]. Cardiomyocyte differentiation promoted by *NKX2-5* is regulated by *BMP* signaling via *SMAD4*, which controls not only the expression but also the nuclear localization of the transcription factor [83].

Environmental factors can determine a CHD, but, for the majority of them, the exact molecular mechanism is not yet fully understood. Lai et al. proposed a mechanism for the association between hyperhomocysteinemia and the development of heart malformations by the downregulation of *NKX2-5* via its interactions with *IGFBP5* [84]. Furthermore, Zhao et al. found a link between maternal gestational diabetes mellitus, single nucleotide polymorphisms of *NKX2-5* and CHD [85].

#### 3.3.4. Genetic Variation Involving *NKX2-5*

*NKX2-5* is one of the most studied genes in relation to heart disease. It appears to be a hypermutable locus, with an estimation that from 2 to 4% of all CHDs can be explained by mutations in the transcription factor, with the majority of them involving the outflow tract and the septal region [75,86]. Table 5 summarizes some of the identified mutations involving genes and their phenotypes.

### 3.4. CRELD1

The Cysteine-Rich with EGF-Like Domains 1 (*CRELD1*) gene encodes for a protein that acts as a cell adhesion and signaling molecule, is highly conserved between species, ranging from *C. elegans* to *Homo sapiens*, and is expressed both during embryogenesis and adult life in a number of different tissues, but mainly the developing heart, limb buds, branchial arches and brain [94].

#### 3.4.1. *CRELD1* in Heart Development

During cardiac embryogenesis, *CRELD1* is expressed in the cells of the atrioventricular canal, endocardial cushions and the outflow tract, being involved in the development of these structures [95]. Knock-out *CRELD1* mice presented fewer cells and less extracellular matrices in the atrioventricular endocardial cushions than wild-type mice [96]. Studies have revealed an interaction between *VEGF*, which seems to act upstream of *CRELD1*, and the calcineurin/*NFATc1* signaling pathway, with *CRELD1* promoting the nuclear translocation of the transcription factor *NFAT*, is known to regulate genes involved in the development of the heart [95,96]. Using a murine model of the spatial inactivation of *CRELD1*, Beckert et al. uncovered that the gene plays an important role not only in the early stages of heart development but also in the maturation and the maintenance of the normal function of the myocardium by regulating transcriptional networks in both atria and ventricles that control the modeling of the extracellular matrix and trabecular formation [97].

#### 3.4.2. Genetic Variation Involving *CRELD1*

*CRELD1* mutations have been described in association with the development of atrioventricular septal defects (AVSD), in both isolated and syndromic patients, with an additional risk in Down syndrome patients, possibly due to the particular genetic background of trisomy 21 [98,99]. It was approximated that between 5 and 10% of the AVSD patients carry a missense mutation in this gene [96]. Table 6 summarizes the pathologic implications of *CRELD1* mutations described so far.

### 3.5. Genetic Variations in Other Transcription Factor and Signaling Molecule Genes

The vascular endothelial growth factor (*VEGF*) family is essential for angiogenesis, and the pathway involving these signaling molecules is also a key regulator of heart morphogenesis [104]. *VEGFA* plays an important role in the early stages of heart development, influencing the formation of the heart tube, and its overexpression can lead to impaired heart tube elongation and looping [105]. In addition, it is involved in the development of the endocardial cushions by mediating the endothelial-to-mesenchymal transition; thus, alterations in the expression of *VEGFA* may lead to atrioventricular septal defects and dysplastic atrioventricular valves [106]. Single nucleotide polymorphisms of *VEGFA*, such as c.-2578C>A, c.+963C>T, c.-634C>G, c.-1154G>A and c.-460T>C, were associated with an increased risk of developing CHDs, especially for tetralogy of Fallot and ventricular septal defects [67,107,108,109]. Moreover, studies have shown that deficiencies in the *VEGF* signaling pathway, due to haploinsufficiency of *VEGF* receptors, also contribute to the pathogenesis of tetralogy of Fallot [110].

The bone morphogenetic protein (*BMP*) signaling pathway is involved in the differentiation of the cardiac mesoderm as it regulates the expression of *NKX2-5* transcription factor [104]. Murine models have uncovered an important role played by *BMP4* in the remodeling and expansion of the endocardial cushions; it is required for the normal septation of the outflow tract and the development of the semilunar valves [111]. This process is mediated by complex molecular interactions between *BMP4* and *TBX1* and by *VEGFA* [112,113]. Although studies have underlined its involvement in cardiac embryogenesis, the relationship between the genetic variation of *BMP4* and CHDs in human populations is still being researched. The association between the single nucleotide polymorphism rs762642 and both atrial and ventricular septal defects was demonstrated in a Chinese population [114,115].

*ISL1* is a transcription factor that characterizes the cardiac progenitor cells [116]. The genetic variants of the promoter of *ISL1* (g.4581A>G, g.4630G>A, g.5085G>A) were associated with the development of tetralogy of Fallot by causing functional changes that alter the transcriptional activity of the promoter, downregulating the transcription factor and its downstream effectors [117]. Additionally, the promoter variants g.4335A>G, g.4477G>A and g.4613G>A were described in patients with ventricular septal defects, demonstrating a similar pathogenetic mechanism [116]. Furthermore, a loss-of-function mutation c.225C>G (p.Tyr75Ter) was identified in a patient presenting with double-outlet right ventricle, altering the transcriptional activity of the gene and the interactions with *TBX20*, another transcription factor involved in heart development [118].

## 4. Conclusions

As previously stated, CHDs are a heterogeneous group of disorders with complex underlying pathologic mechanisms. In one study, it was estimated that over 400 genes are likely to be involved in their etiology [119]. In addition to genetic mutations, the researchers’ focus was also directed towards common and non-coding variants, as well as epigenetic factors and interactions with environmental contributors, as an effort to uncover some of the missing heritability [104,119].

Understanding new concepts regarding heart development has led to a paradigm shift in the pathogenesis of CHDs, from the multifactorial inheritance hypothesis to the “one heart disease—several mechanisms—several genes” hypothesis. It states that having a heart disease may be caused by errors in several embryological mechanisms in a developmental sequence, each of which are orchestrated by a specific set of genes, similarly to the “off targets” model from cancer research [119,120]. This paradigm shift has practical implications by changing the focus from the anatomical defect to the developmental mechanism. In order to give adequate genetic advice, it is mandatory to describe the cardiac anatomy accurately so that the suggestive embryological pathways can be identified and candidate genes can be determined [120]. Providing a molecular diagnosis of CHDs is important not only for future family planning but also for effective future patient management, due to insights in the evolution of the disorders (for example, an association with conductance disturbances, cardiomyopathies or extracardiac symptoms), as well as the outcome after surgery or other treatments (some genetic variations have been described as risk factors for morbidity and mortality after surgery) [119].

Transcription factors and signaling molecules are promising candidates for studies regarding the genetic burden of CHD. Although many of the pathogenetic mechanisms surrounding heart development have been described, further research is needed in order to uncover their downstream targets and the complex network of interactions with non-genetic risk factors for a better molecular–phenotype correlation.

## Figures and Tables

**Table 1 children-10-00812-t001:** The contribution of different etiologic factors in the development of CHD (adapted after [5,6,7]).

Non-Genetic Contribution~10%	Genetic Contribution~90%
Convincing evidence - Severe maternal obesity Highly suggestive evidence - Antidepressants (lithium) - Maternal obesity - Maternal alcohol consumption - Maternal fever in the first trimester Suggestive evidence - Antidepressants (SSRIs) - Paternal advanced age - Paternal smoking Other - Retinoic acid - Anticonvulsants - Antiarrhythmics (class III) - Maternal diabetes mellitus - Maternal viral infections (rubella)	Aneuploidy (~13%) - Trisomy 21, 13 and 18 - Monosomy X CNV (~10%) - Del22q11.2 - Del7q11.23 - Del5p15.2 - Del11q - Del8p23.1 - Del1q21.1 - Dup22q11 De novo mutations (~8%) - Chromatin-modifying genes Transmitted mutations (~3%) - Transcription factors genes - Signaling molecules genes - Structural genes Unknown (~56%)

CNV, Copy number variation; SSRI, selective serotonin reuptake inhibitor.

**Table 2 children-10-00812-t002:** The milestones of cardiac development and the events leading to congenital heart defects (adapted with permission from [11]. 2013 Wiley Periodicals, Inc., Hoboken, NJ, United States).

Carnegie Stage	Age (DPC)	Events	CHD
CS8	17–19	Development of the heart-forming region	
CS9	19–21	Embryonic folding and placement of the heart-forming region in the final position	
CS10	22–23	Formation of the heart tube	
		Looping	
		Ventricular ballooning	
CS11	23–26	Atrial ballooning	
CS12	26–30	Formation of the primary atrial septum	
		Development of the muscular part of the ventricular septum	Muscular VSD
CS13	28–32	Formation of the atrioventricular cushions	
		Attachment of the pulmonary veins to the left atrium	
CS14	31–35	Appearance of the outflow tract ridges	
CS15	35–38	Formation of the secondary foramen	
		Beginning of the septation of the outflow tract	TA, TGA, TOF
		Migration of the neural crest cells to the outflow tract	DORV, pulmonary atresia
CS16	37–42	Closure of the ostium primum	AVSD, ASD I
		Outflow ridges approaching the ventricular septum	
CS17	42–44	Formation of the secondary atrial septum	ASD II
		Separation of the atrioventricular communication	
		Completion of the outflow tract septation	Membranous VSD
CS18	44–48	Formation of the atrioventricular valves	Tricuspid atresia

ASD I, Atrial septal defect type ostium primum; ASD II, atrial septal defect type ostium secundum; AVSD, atrioventricular septal defect; CS, Carnegie stage; DORV, double-outlet right ventricle; DPC, days post-conception; TA, truncus arteriosus; TGA, transposition of great arteries; TOF, tetralogy of Fallot; VSD, ventricular septal defect.

**Table 3 children-10-00812-t003:** Genotype–phenotype correlation in *TBX5* mutations.

Mutation Type	Location	Cardiac Phenotype	Other Phenotypes	Reference
Nonsense—LOF	c.577G>T (p.Gly193Ter)	TAPVR and ASD	Triphalangeal thumb	[41]
Missense	c.322C>A (p.Pro108Thr)	Tricuspid atresia	-	[42]
Compound (missense + nonsense)	c.791G>A + c.835C>T (p.Arg264Lys + p.Arg279Ter)	HLHS, VSD and PDA	Hypoplastic thumb	[43]
Missense—GOF	c.373G>A (p.Gly125Arg)	Early onset paroxysmal AF, ASD and VSD	(Sub)luxation of the radial head, carpal synostosis, scoliosis and scapular dysplasia	[44]
Missense—LOF	c.668C>T (p.Thr223Met)	Long QT syndrome, ASD and VSD	-	[45]
Nonsense—LOF	c.835C>T (p.Arg279Ter)	DCM, bicuspid aortic valve, and first-degree AV block	Hypoplastic thumb	[46]
Missense—LOF	c.710G>A (p.Arg237Gln)	DCM, ASD, sick sinus syndrome and AF	Mild pectus excavatum	[46]
Microdeletion + microinsertion	c.627delinsGTGACTCA GGAAACGCTTTCCT GA	ASD and VSD	Bilateral dysplasia of radius and thumb	[47]
Intragenic duplication	Dup12q24.1—11 kb (exons 1–6)	ASD, VSD and complete AV block	Bilateral dysplasia of radius and thumb	[48]
Intragenic duplication	Dup12q24.21—48 kb (exons 2–9)	AVSD, pulmonary stenosis, HLHS, atrial flutter, AF and sick sinus syndrome	Bilateral ulnar hypoplasia, syndactyly and fifth finger clinodactyly	[49]
Contiguous deletion	Del12q24.13-q24.21—3.1 Mb (including *TBX5* and *TBX3*)	ASD, VSD, right pulmonary artery hypoplasia and high-grade second-degree AV block	Bilateral dysplasia of radius and thumb, fifth finger clinodactyly, absent nipples bilaterally, cryptorchidism and glandular hypospadias	[50]
Contiguous duplication	Dup12q24.21—399 kb (including *TBX5* and *TBX3*)	ASD, PDA, aortic stenosis, bicuspid aortic valve and AF	Absence of distal inter phalangeal joints of the thumb, hypoplastic thenar eminence, camptodactyly and supernumerary nipple	[51]

AF, Atrial fibrillation; ASD, atrial septal defect; AV block, atrioventricular block; AVSD, atrioventricular septal defect; DCM, dilated cardiomyopathy; GOF, gain-of-function; HLHS, hypoplastic left heart syndrome; LOF, loss-of-function; PDA, patent ductus arteriosus; TAPVR, total anomalous pulmonary venous return; VSD, ventricular septal defect.

**Table 4 children-10-00812-t004:** Genotype–phenotype correlation involving *GATA4*.

Mutation Type	Location	Cardiac Phenotype	Other Phenotypes	Reference
Missense	c.431C>T (p.Ala144Val)	Pulmonary atresia and ASD	-	[42]
Missense	c.487C>T (p.Pro163Ser)	ASD	Sexual development disorder	[66]
Missense	c.886G>A (p.Gly296Ser)	VSD	-	[67]
Missense	c.929T>C (p.Met310Thr)	ASD, AVSD, pulmonary stenosis, AF, paroxysmal VT and junctional premature beat with aberrant ventricular conduction	-	[68]
Synonymous	c.99G>T (p.Ala33Ala)	ASD, VSD, coarctation of aorta and TOF	-	[69]
Synonymous	c.822C>T (p.Cys274Cys) c.906C>T (p.His302His)	Bicuspid aortic valve	-	[70]
SNP (regulatory variant)	g.31360T>C g.31436G>A g.31437C>A g.31487C>G g.31856C>T	ASD	-	[71]
Insertion	Ins4+5G>A	TOF	-	[42]
Microdeletion	Del8p23.1	AVSD	Psychomotor delay	[42]
Deletion	DelGATA4	VSD, ASD and pulmonary stenosis	Craniofacial dysmorphism and ectopic kidney	[72]
Deletion	Del8p23.1—between 2.945 and 6.352 Mb	DORV, AVSD, pulmonary atresia, TGA and ASD	Congenital diaphragmatic hernia and craniofacial dysmorphism	[73]

AF, Atrial fibrillation; ASD, atrial septal defect; AVSD, atrioventricular septal defect; DORV, double-outlet right ventricle; TGA, transposition of great arteries; TOF, tetralogy of Fallot; VSD, ventricular septal defect; VT, ventricular tachycardia.

**Table 5 children-10-00812-t005:** Genotype–phenotype correlation involving *NKX2-5* gene.

Mutation Type	Location	Cardiac Phenotype	Other Phenotypes	Reference
Missense	c.182C>G (p.Ala61Gly)	VSD and coarctation of aorta	-	[87]
Missense	c.284G>T (p.Arg95Leu)	VSD	-	[87]
Missense	c.391G>A (p.Glu131Lys)	ASD and TOF	-	[87]
Missense	c.443C>A (p.Ala148Glu)	DORV, TGA, ASD, VSD, coarctation of aorta and atretic aortic valve	-	[87]
Missense	c.739C>G (p.Pro247Ala)	TOF	-	[87]
Missense	c.413G>A (p.Arg138Glu)	Familial ASD	-	[88]
Missense	c.561G>C (p.Gln187His)	Familial ASD	-	[88]
Missense	c.355G>T (p.Ala119Ser)	HLHS	-	[89]
Synonymous	Exon 2	Septal defects	-	[90]
Nonsense	c.541C>T (p.Gln181Ter)	ASD and AV block	-	[91]
Nonsense	c.574A>T (p.Lys192Ter)	Bicuspid aortic valve	-	[92]
Frameshift	c.397-400del	DORV, AVSD and TAPVR	Extracardiac heterotaxy syndrome features	[93]

ASD, Atrial septal defect; AV block, atrioventricular block; AVSD, atrioventricular septal defect; DORV, double-outlet right ventricle; HLHS, hypoplastic left heart syndrome; TAPVR, total anomalous pulmonary venous return; TGA, transposition of great arteries; TOF, tetralogy of Fallot; VSD, ventricular septal defect.

**Table 6 children-10-00812-t006:** Genotype–phenotype correlation involving *CRELD1* gene.

Mutation Type	Location	Cardiac Phenotype	Other Phenotypes	Reference
Missense	c.985C>T (p.Arg329Cys)	AVSD, VSD and PPHT	-	[100,101]
Missense	c.932C>T (p.Thr311Ile)	AVSD	-	[100]
Missense	c.320G>A (p.Arg107His)	AVSD, pulmonary atresia, dextrocardia and right aortic arch (heterotaxy syndrome)	-	[100]
Missense	c.973G>A (p.Glu325Lys)	AVSD	Down syndrome	[98,102]
Missense	c.857C>G (p.Pro286Arg)	AVSD	-	[102]
Synonymous	c.1628G>A (p.Lys336Lys)	ASD	-	[103]
SNP	c.-103G>T (5′UTR)	ASD	-	[103]
SNP	c.1048+23G>T (intronic)	ASD	-	[103]

ASD, Atrial septal defect; AVSD, atrioventricular septal defect; PPHT, persistent pulmonary hypertension; SNP, single nucleotide polymorphism; UTR, untranslated region; VSD, ventricular septal defect.

## Data Availability

Not applicable.

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
