# Peer review of "Genetic Alterations of Transcription Factors and Signaling Molecules Involved in the Development of Congenital Heart Defects—A Narrative Review"

_children, 2023, doi:10.3390/children10050812_

Round 1

Reviewer 1 Report

This review is well written, but fails to adequately address the topic cited by the authors, which is to review genetic factors related to heart development and defects. For example, the entire first half of the manuscript attempts to describe normal heart development. This description is not essential to understanding the genetic abnormalities described in the second half. There are also some mistakes made in this description. For example, formation of the right atrium is by incorporation of the right horn of the sinus venosus not the right common cardinal vein. As another example, the heart is suspended from the gut tube by the dorsal mesentery not from the body wall. Yet another involves the description of the foramen ovale. The foramen is not the portion of the primary septum that remains, but rather is the oblique passageway between the right and left atria created by the overlapping primary and secondary septa. The primary septum ultimately forms the valve of the foramen ovale.

Another issue involving the description of normal heart development is that reference #8 is cited at least 17 times, which may represent giving credit where credit is due, but suggests that the authors have not adequately reviewed the literature, but have depended too much upon one source for their report.

Another question arises as to why the authors lump all vascular defects with heart defects. Why is transposition of the great vessels considered a heart defect when it clearly involves blood vessels and not the heart. And how in the world can a patent ductus arteriosus be a heart defect. It would seem that in this molecular era that we should do better at defining and classifying defects that are not related, such as heart versus vascular abnormalities, in order to find relevant mechanisms for congenital abnormalities involving the two subjects. 

The strongest part of the manuscript is the discussion of genes involved in heart development and the origin of some abnormalities. I believe this section could be expanded and the various correlations between genetic abnormalities and heart and vascular defects and gene gene interactions described in more detail to provide a relevant review of the subject.

Author Response

We considered that a description of the normal heart development would be useful in relation to the specific role played by each of the transcription factors and signaling molecules further described. We addressed the issues specifically mentioned regarding the development, thank you for pointing them out. 

Regarding the classification of congenital heart defects, we used the one provided by Botto et all. because we considered it the most relevant, being based on embryological and anatomical criteria. Transposition of the great arteries was included in the classification due to the origin from abnormal outflow tract development (the lack of counterclockwise rotation and abnormal septation), leading to abnormal ventricle-to-arterial connections (1,2).

(1) Narematsu M, Kamimura T, Yamagishi T, Fukui M, Nakajima Y. Impaired development of left anterior heart field by ectopic retinoic acid causes transposition of the great arteries. J Am Heart Assoc. 2015 Apr 30;4(5):e001889. doi: 10.1161/JAHA.115.001889. PMID: 25929268; PMCID: PMC4599416.

(2) Nakajima Y. Mechanism responsible for D-transposition of the great arteries: Is this part of the spectrum of right isomerism? Congenit Anom (Kyoto). 2016 Sep;56(5):196-202. doi: 10.1111/cga.12176. PMID: 27329052.

Reviewer 2 Report

This review provides an overview of the current knowledge regarding some of the genetic mechanisms involved in the embryological development of the heart, and the association between genetic variations in transcription factors and signaling molecules involved in congenital heart defects, hence is interesting.

Regarding the association of the variations in the genes encoding transcription factors with congenital heart defects, other relevant articles should be referred to and simply discussed in the last paragraph: 3.5. Genetic variations in other transcription factor genes.

Author Response

Thank you very much for the appreciation. We added the suggested section.

Reviewer 3 Report

This a well written comprehensive narrative review. I have the following suggestions:

The introduction section provides a comprehensive overview of congenital heart defects (CHD), their prevalence, etiology, and classification. However, there are some aspects that can be improved or further elaborated upon to enhance the clarity and coherence of the section.

The authors mention the classification of CHD by Botto et al. but do not provide the citation. Please provide the appropriate reference for this classification.

In lines 39-47, the authors discuss the prevalence of CHD, noting an increase in the prevalence of "mild" lesions while "severe" lesions have remained relatively constant. It would be helpful to provide a brief explanation of the difference between "mild" and "severe" lesions for readers who may not be familiar with these terms.

In lines 49-60, the authors discuss environmental and genetic factors contributing to CHD. It might be helpful to provide some examples of gene-to-gene and gene-to-environment interactions that are currently being researched to provide context for readers.

In lines 65-69, the authors provide the aim of the review. It would be helpful to provide a brief outline of the structure of the review to help guide the reader through the manuscript.

In lines 71-124, the authors provide a detailed description of the embryologic stages of heart development. However, this section is quite dense and can be difficult to follow for readers who are not familiar with the subject. To improve readability, the authors could consider breaking down the text into shorter paragraphs or using subheadings to indicate different stages of development.

In the manuscript, the authors mention various transcription factors (e.g., Nkx2.5, Islet1, TBX) and signaling molecules (e.g., Wnt, BMP) involved in heart development. It would be helpful to provide a brief explanation of the roles these factors play in the process to provide context for readers who may not be familiar with their function.

The second section of the manuscript provides a detailed account of transcription factors and signaling molecules involved in the development of congenital heart defects (CHD). Overall, the section is informative and well-organized, although a few suggestions can be made to improve clarity and readability.

For all sections including TBX5 and GATA4 sections, consider breaking the text into subheadings to enhance readability and make it easier for the reader to follow the specific roles played by these transcription factors in CHD. Examples of subheadings could be "TBX5 in heart and limb development," "TBX5 in the ventricular conduction system," "Gene-environment interactions involving TBX5," and "GATA4 in cardiomyocyte differentiation" and "GATA4 in the development of the atrioventricular region."

In the GATA4 section, lines 248-252, the authors mention various in vitro studies. Providing specific references for these studies would strengthen the credibility of the manuscript.

Author Response

Thank you for your appreciation. We addressed the specific issues pointed out in the revised version. Also, we provided subheadings for the denser parts of our review in order to enhance readability, as you suggested.